biomathematics/biomedical engineering

wavelets, machine learning, resuscitation

**Author for correspondence:**
Diya Sashidhar
e-mail: dsashid@uw.edu

This work was supported in part by the American Heart Association under Grant 19SFRN34930005 and the Washington Research Foundation.

# Machine learning and feature engineering for predicting pulse presence during chest compressions

Diya Sashidhar[1,2], Heemun Kwok[2,3], Jason Coult[2,5], Jennifer Blackwood[2], Peter J. Kudenchuk[2,4], Shiv Bhandari[2,5], Thomas D. Rea[2,5] and J. Nathan Kutz[1,2]

[1]Department of Applied Mathematics, [2]Center for Progress in Resuscitation, [3]Department of Emergency Medicine, [4]Division of Cardiology, Department of Medicine, and [5]Division of General Internal Medicine, Department of Medicine, University of Washington, Seattle, WA 98195, USA

DS, 0000-0003-2914-5791; JC, 0000-0003-0568-0968; JNK, 0000-0002-6004-2275

Current resuscitation protocols require pausing chest compressions during cardiopulmonary resuscitation (CPR) to check for a pulse. However, pausing CPR when a patient is pulseless can worsen patient outcomes. Our objective was to design and evaluate an ECG-based algorithm that predicts pulse presence with or without CPR. We evaluated 383 patients being treated for out-of-hospital cardiac arrest with real-time ECG, impedance and audio recordings. Paired ECG segments having an organized rhythm immediately preceding a pulse check (during CPR) and during the pulse check (without CPR) were extracted. Patients were randomly divided into 60% training and 40% test groups. From training data, we developed an algorithm to predict the clinical pulse presence based on the wavelet transform of the bandpass-filtered ECG. Principal component analysis was used to reduce dimensionality, and we then trained a linear discriminant model using three principal component modes as input features. Overall, 38% (351/912) of checks had a spontaneous pulse. AUCs for predicting pulse presence with and without CPR on test data were 0.84 (95% CI (0.80, 0.88)) and 0.89 (95% CI (0.86, 0.92)), respectively. This ECG-based algorithm demonstrates potential to improve resuscitation by predicting the presence of a spontaneous pulse without pausing CPR with moderate accuracy.

**Figure 1.** Overview of algorithmic architecture for real-time pulse presence classification. Time series from ECG data is transformed to a feature space generated from wavelet analysis. Dominant correlated features in this space are used for assessment of pulse presence.

# 1. Introduction

Machine learning (ML) and artificial intelligence (AI) algorithms are transforming the scientific landscape [1,2]. From self-driving cars and autonomous vehicles to digital twins and manufacturing, there are few scientific and engineering disciplines that have not been profoundly impacted by the rise of machine learning and AI methods. Medicine is no exception, with a significant growth of machine learning and AI methods developed for applications ranging from imaging [3,4] to personalized medicine [5]. We specifically developed a novel ECG-based ML algorithm that demonstrates the potential to improve resuscitation by predicting presence of a spontaneous pulse without pausing cardiopulmonary resuscitation (CPR).

ECG data and AI/ML algorithms are ideally poised to better inform resuscitation efforts in patients with cardiac arrest. Hundreds of thousands of people suffer an out-of-hospital cardiac arrest (OHCA) each year [6]. Patients who suffer cardiac arrest require time-critical, life-saving interventions for successful resuscitation. These interventions include CPR, which is a combination of chest compressions, ventilations and advanced airway management. Additional interventions, such as electrical defibrillation and drug administration, depend upon knowledge of the cardiac rhythm and the presence of a pulse. Ideally, the rhythm and presence of a mechanical pulse would be monitored continuously during resuscitation to help guide treatment decisions. However, chest compressions preclude accurate assessment of rhythm or a *spontaneous* pulse, because compressions cause artefact in the ECG and can generate some measure of pulse themselves regardless of the underlying rhythm. Conversely, pausing CPR to assess these parameters comes at the cost of depriving the patient, if actually pulseless, of needed haemodynamic support and can adversely affect outcome. Although these pauses are intended to be brief in duration, interruptions in CPR to evaluate rhythm and pulse may take up to 20 s or longer, with increasing duration inversely associated with chances of survival [7,8]. Thus, current guidelines reflect a balance between interrupting CPR to manually assess a patient's pulse presence versus performing continuous chest compressions in recommending an interruption in chest compressions for rhythm and pulse assessment only once every 2 min [9,10]. The ability to detect a spontaneous pulse in real time during ongoing CPR would better inform care providers of the patient's clinical status throughout resuscitation and afford more time-sensitive, better-informed treatment decisions to achieve improved clinical outcomes.

In the current study, we developed a scalable approach that integrates automated feature selection with various machine learning techniques (figure 1). This approach uses wavelet transforms followed by principal component analysis, allowing for the potential capture of subtle ECG morphologies while reducing feature dimensionality. Using only three dimensions of the reduced wavelet transform, we evaluated a diverse number of ML models, including deep learning architectures, that predict the pulse presence of organized rhythm segments both with and without CPR. We hypothesized that this approach would provide accurate detection of spontaneous pulse during ongoing CPR.

# 2. Methods

## 2.1. Description of data

This retrospective observational study evaluated a convenience sample of 383 patients who experienced OHCA presenting with one of the following rhythms: ventricular fibrillation (VF), pulseless electrical

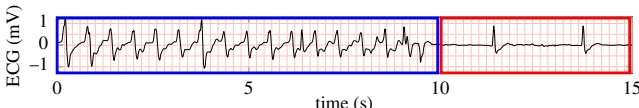

**Figure 2.** ECG after using 4-tap Butterworth filter process. A 10-s segment during CPR is highlighted in blue (left), and a 5-s segment during CPR pause for pulse check is highlighted in red (right).

activity (PEA) or asystole, and underwent attempted resuscitation by emergency medical services (EMS) in King County, WA between 2006 and 2014 and had an available electronic defibrillator recording. While patients may have had an initial rhythm of VF, PEA or asystole, all patients achieved an organized rhythm at some point. An organized rhythm is defined as discrete repetitive electrical complexes on the ECG (indicative of ventricular depolarization) at a rate that would be expected to produce a discernible pulse, but may or may not do so. Only organized clips were used to train and test the algorithm in study. EMS protocols followed American Heart Association guidelines and included the provision of manual CPR [10]. The defibrillator recording from each case included continuous measurement of the ECG and thoracic impedance, and an audio channel containing EMS narration of events during the course of resuscitation. Each defibrillator recording was reviewed by experts to identify 2 min rhythm or pulse checks from the impedance signal and to annotate the rhythm at each check [11–13]. Organized rhythms were further categorized as either mechanical pulse present or absent (i.e. pulseless electrical activity) by reviewing the audio recording, as EMS rescuers verbalize the results of manual pulse assessment and confirm any positive pulse assessment by blood pressure measurement per protocol [14]. The study was reviewed and approved by the Human Subjects Review Committee of the University of Washington which granted exemption from informed consent under established minimal risk criteria in compliance with applicable government regulations (STUDY00013007), with concurrence by the Research Oversight Review Committee of Public Health–Seattle and King County who ceded their approval to the University of Washington.

Patients were randomized into training (60% of patients) and test (40% of patients) groups for algorithm development. Data was randomized on a patient level in order to test generalizability to patients in a withheld test set. We developed and evaluated the algorithm on adjacent ECG segments at each pulse assessment with an organized rhythm. Specifically, for each pulse check, a 10 s ECG segment was collected during the period of ongoing CPR just prior to the pulse check, immediately followed by a 5 s adjacent segment collected without CPR that was collected during the pulse check. We assumed pulse presence was unlikely to have changed during this very short interval (figure 2). The number of pulse checks depended upon the course of resuscitation and was therefore variable across patients. For example, a patient who achieved a sustained spontaneous pulse quickly may have had only one pulse check, while another patient who did not ultimately survive may have had many pulse checks. To prevent the over-representation of data from any single patient, we set a maximum of three randomly selected 15 s periods from each category (*Pulse* or *Pulseless*) per patient. Table 1 also shows the median number of samples (including pulse and pulseless) for each patient for training and test sets, respectively.

## 2.2. Data preprocessing and noise removal

Data were collected from MRx and Forerunner 3 (Philips Healthcare, Bothell, WA), and Lifepak 12 and Lifepak 15 (Physio-Control, Redmond, WA) automated biphasic defibrillators, with ECG sampling rates ranging from 125 to 250 Hz. To ensure that the underlying method would be applicable across this range in sampling rates, all data were low-pass filtered at 40 Hz after resampling all data to a common sampling rate of 250 Hz.

Since the lowest original sampling rate is 125 Hz (with a Nyquist frequency of 62.5 Hz), we assume that the 40 Hz cut-off frequency assures the same maximum frequency content across all devices, eliminating the possibility that the algorithm would use frequencies above 40 Hz that may not be as well-resolved in devices with lower sampling rates.

We applied filtering to remove high-frequency electrical noise and low-frequency drift using a fourth-order Butterworth bandpass filter from 1 to 40 Hz implemented with a forwards–backwards implementation to preserve linear phase. Of note, this filter was not designed to remove artefact from CPR, which is concentrated primarily at the frequency of manual chest compressions (2 Hz) but can exhibit harmonic artefact (e.g. up to 20 Hz) that can overlap with underlying heart activity [15,16].

**Table 1.** Patient demographics. Note: significance was calculated using Mann–Whitney $U$ test for continuous variables and $\chi^2$ for proportions.

| patient characteristics | training ($N = 230$) | test ($N = 153$) | all ($N = 383$) | significance |
|---|---|---|---|---|
| initial rhythm | | | | |
| ventricular fibrillation (VF) | 192 (83.5) | 128 (83.7) | 320 (83.6) | 0.96 |
| asystole | 11 (4.7) | 5 (2.2) | 16 (4.2) | 0.47 |
| pulseless electrical activity (PEA) | 26 (11.3) | 20 (13.1) | 46 (12.0) | 0.6 |
| indeterminate | 1 (0.4) | 0 (0.0) | 1 (0.2) | * |
| female, $n$(%) | 63 (27.4) | 42 (27.5) | 105 (27.4) | 0.72 |
| age, median (IQR) | 64 (53, 74.5) | 62 (50.5, 73) | 63 (52, 74) | 0.62 |
| cardiac etiology, $n$(%) | 201 (87.4) | 128 (83.7) | 329 (85.9) | 0.37 |
| location, $n$(%) | | | | |
| home | 155 (67.4) | 99 (64.7) | 254 (66.3) | 0.53 |
| public | 61 (26.5) | 43 (28.1) | 104 (27.2) | 0.84 |
| nursing home | 13 (5.7) | 12 (7.8) | 25 (6.5) | 0.4 |
| arrest before EMS arrival, $n$(%) | 215 (92.6) | 141 (92.2) | 356 (93) | 0.73 |
| witnessed, $n$(%) | 164 (71.3) | 100 (65.4) | 264 (68.9) | 0.19 |
| bystander CPR, $n$(%) | 151 (65.7) | 97 (63.4) | 248 (64.8) | 0.52 |
| EMS response (min), median (IQR) | 5.1 (4.3, 6.9) | 5.2 (4.4, 6.8) | 5.2 (4.4, 6.9) | 0.47 |
| total shocks, median (IQR) | 3 (1, 7) | 2 (1, 3.5) | 2 (1, 5) | 0.06 |
| return of spontaneous circulation, $n$(%) | 156 (67.8) | 109 (71.2) | 265 (69.2) | 0.58 |
| admit to hospital, $n$(%) | 164 (71.3) | 110 (71.9) | 274 (71.5) | 0.42 |
| survive to hospital discharge, $n$(%) | 108 (47) | 69 (45.1) | 177 (46.2) | 0.51 |
| median number of samples per patient | 2.0 | 2.5 | 2.0 | — |

Other studies have attempted to reduce CPR artefact using adaptive filters, Kalman filters and notch filters [15,17].

By contrast, rather than attempt to remove CPR artefact via filtering as in prior studies [15], the current method employs a combination of time-frequency analysis and machine learning to allow data-driven exclusion of frequencies confounded by CPR artefact.

## 2.3. Algorithm development

### 2.3.1. Wavelet transforms

We used wavelets as a decomposition method for analysis and representation of time-frequency signals [18,19]. As the mathematical structure for constructing the multi-resolution analysis of data such as the ECG time series or spatial data such as images [19,20], wavelets are foundational in the representation of data that have multi-scale spatio-temporal features. Leading image compression technologies such as JPEG 2000 are wavelet-based encoding schemes which leverage multi-resolution analysis to sparsely represent important image features for the compression process [21]. Importantly, wavelets provide a recursive architecture which extracts dominant time-frequency features within prescribed time-bins, much like the windowed Fourier transform or Gabor transform [20].

In the current investigation, the wavelet transform was used to analyse the time-frequency structure of the filtered ECG by constructing the ubiquitous scalogram, which is equivalent to the spectrogram created using Gabor transforms [20]. This scalogram provided a visual representation of the time and frequency content of the ECG data and a critical feature space to extract meaningful portions of the signal that may predict pulse.

The wavelet transform was computed by convolving a wavelet function with the time-series recording from the ECG. The wavelet window, whose temporal width is recursively cut in half (or

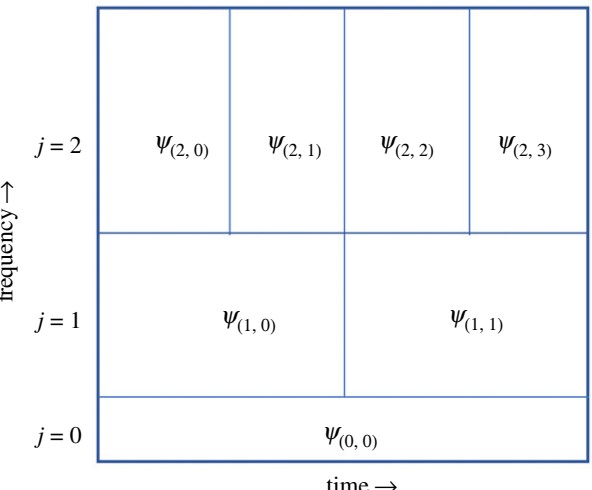

**Figure 3.** Wavelet transform template.

doubled) hierarchically, was translated across the entire ECG time series as its time-frequency information was extracted (figure 3). The wavelet is represented by the functional form

$$\Psi_{j,k}(t) = \frac{1}{a_j} \Psi\left(\frac{t - b_k}{a_j}\right), \tag{2.1}$$

where $a_j$ denotes the wavelet dilation parameter, $b_k$ is the centre position of the wavelet window, $t$ is the current time point and $\Psi$ is the mother wavelet. The parameter $b_k$ is chosen so that the wavelet window translates across the entire time history of the ECG data segment (figure 4). The wavelet transform convolves (2.1) with the ECG signal $f(t)$ and gives a new representation that is parametrized by both $a_j$ and $b_k$

$$W_\Psi(f)(a_j, b_k) = \int_0^T f(t)\bar{\Psi}_{j,k}(t)\mathrm{d}t, \tag{2.2}$$

where the ECG signal is recorded for $t \in [0, T]$. The value of $W_\Psi(f)(a_j, b_k)$ for each $a_j$ and $b_k$ gives the spectral content, or energy, at a specific time-frequency location. By plotting the entire time-frequency plane, the scalogram is constructed from equation (2.2). Specifically, the energy of a signal ($E$) in a window corresponding to dilation parameter $a$ and centred at $b$ is given by

$$E(a, b) = |W_\Psi(f)(a, b)|^2. \tag{2.3}$$

To calculate the scalogram, we integrated these values across all discrete values of $a_j$ and $b_k$ used when discretizing the time-series signal.

Since wavelets can extract localized features in the time-frequency domain, we applied the wavelet transforms to the detrended ECG time-series measurements (figure 5).

Depending on the wavelet type and sifting window size, the wavelet transform can capture both the morphology of sharp peaks such as the QRS wave as well as the nuances of the $P$ wave. The choice of the wavelet depends on the nature of the signal. Since ECGs are periodic and have oscillatory bursts as seen in QRS complexes, we chose the bump wavelet for extraction of our time-frequency features. It should be noted that out of 14 different wavelets, the bump wavelet had the highest model performance (electronic supplementary material, table S3), thus justifying this choice.

The oscillatory shape of bump wavelet enables it to capture both the morphology and sharpness of the QRS wave and the dome-like structures of the $P$ and $T$ waves. Importantly, the wavelet transform can produce scalograms which will be used as the feature space for identifying the pulse presence of the heart.

### 2.3.2. Principal component analysis

The scalograms produced using the wavelet transforms were typically high-dimensional (pixel space) images, making it difficult to both process and directly identify key signatures in the data. We used *principal component analysis* (PCA) to reduce the dimensionality of the data and to identify dominant

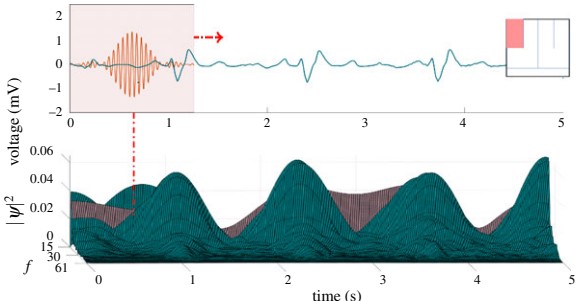

**Figure 4.** Scalogram is calculated by convolving the bump wavelet with the filtered ECG as the window translates across the time.

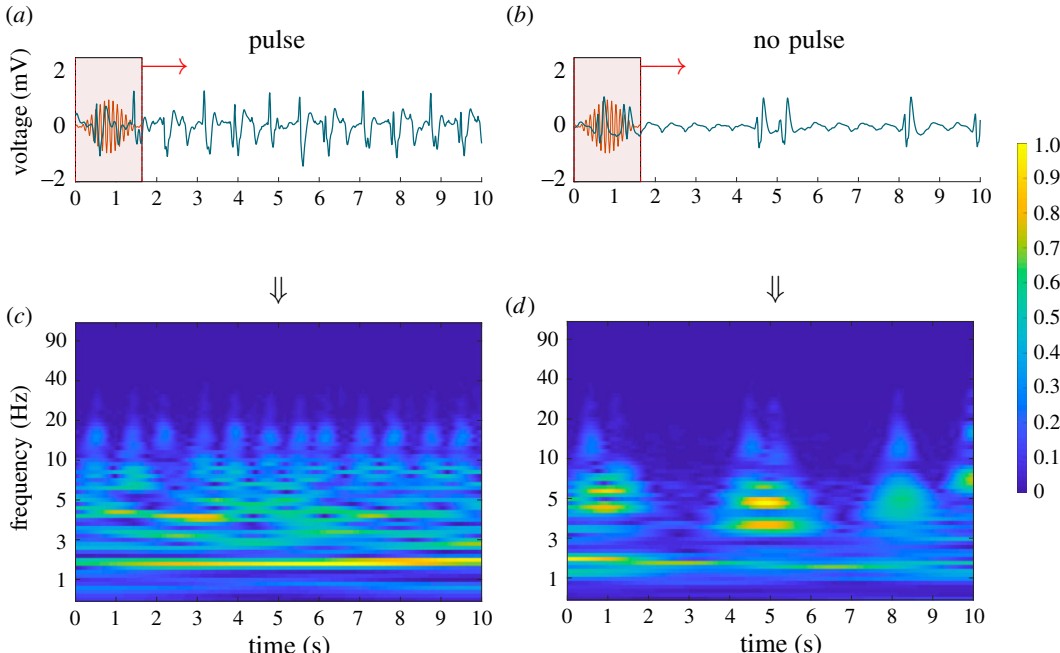

**Figure 5.** (a) Application of bump wavelet to segment of ECG (during CPR) with Pulse ROSC label, (b) Application of bump wavelet to segment of ECG (during CPR) with No Pulse label, (c) Scalogram generated by convolving various scales of bump wavelet with respective ECG segment with Pulse ROSC label, (d) Scalogram generated by convolving various scales of bump wavelet with respective ECG segment with No Pulse ROSC label series.

features in the scalograms. To help illustrate this concept, we first introduce the *singular value decomposition* (SVD) on which PCA is based.

Suppose there is a matrix $A \in \mathbb{C}^{m \times n}$ with $n$ columns of vectorized scalograms ($a_i$ for $i = 1, \dots, n$) of size $m$, where $m \gg n$. The SVD can be used to decompose this single matrix into a product of three matrices.

$$\mathbf{A} = \mathbf{U}\mathbf{\Sigma}\mathbf{V}^T$$

$$\begin{bmatrix} \vdots & \vdots & \vdots \\ a_1 & & a_n \\ \vdots & \vdots & \vdots \end{bmatrix} = \begin{bmatrix} \vdots & \vdots & \vdots \\ u_1 & & u_n \\ \vdots & \vdots & \vdots \end{bmatrix} \begin{bmatrix} \sigma_1 & & 0 \\ 0 & \ddots & 0 \\ & 0 & \sigma_n \end{bmatrix} \begin{bmatrix} \vdots & \vdots & \vdots \\ v_1 & & v_n \\ \vdots & \vdots & \vdots \end{bmatrix}^T,$$

where $\mathbf{U} \in \mathbb{C}^{m \times n}$ is a unitary matrix, $\mathbf{\Sigma} \in \mathbb{C}^{n \times n}$ is a diagonal matrix, and $\mathbf{V} \in \mathbb{C}^{n \times n}$ is a unitary matrix.

The columns of $\mathbf{U}$ and $\mathbf{V}$ are known as the left and right singular vectors, respectively. The columns can be thought of as 'modes' of the system, where each mode represents a dominant feature in the data. Thus, this decomposition has the ability to rank and analyse each feature produced by correlations in the data. Specifically, the left singular vectors, or the spatial modes when reshaped into matrices, show specific regions in the scalogram that exhibit the most variance. The right singular vectors, or the

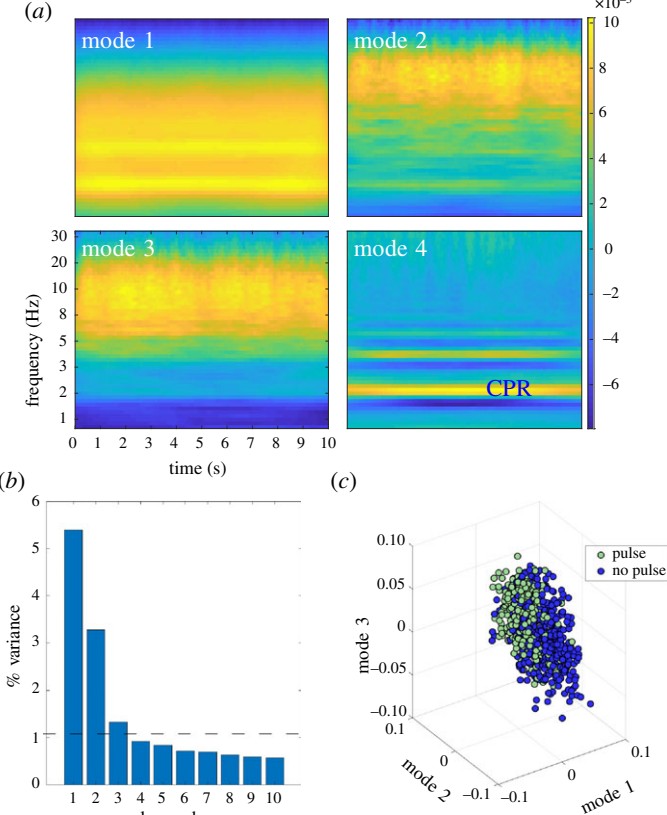

**Figure 6.** (*a*) First four spatial modes of the system of reshaped scalograms (during CPR). (*b*) Normalized singular values of the system of reshaped scalograms. The dotted line represents the variance cut-off. (*c*) Various combinations of temporal modes of system of reshaped scalograms.

temporal modes, show how each scalogram projects onto these modes. Thus the SVD ultimately provides a feature engineering tool which allows for clustering and classification tasks.

The matrix $\Sigma$ comprises descending diagonal elements called singular values. These singular values, which are associated with the columns of **U** and **V**, capture the variance of the data in each principal direction. In other words, the SVD creates a hierarchy of dominant features and computes mode 'weights' using singular values.

This hierarchy helps create a low-rank approximation of the data, $\tilde{A}$. In other words, the first $r$ modes can produce the best low-rank feature representation of the data in a least-squares sense.

$$\tilde{A}_{m\times r} \approx \tilde{\mathbf{U}}_{m\times r}\tilde{\Sigma}_{r\times r}\tilde{\mathbf{V}}^{T}_{r\times r}. \tag{2.4}$$

This gives a significantly reduced matrix that is easier to manipulate and interpret. This concept of SVD is the foundation of PCA. Specifically, PCA involves taking the SVD of matrix $A$ with normalized and centred data columns [20].

We considered a maximum of 10 PCA modes for inclusion as classification features, and the choice of $r$ involved a balance of accuracy and simplicity. As there was only a marginal increase in performance by the successive addition of modes, we chose the Pareto optimal solution by picking the more parsimonious model [20]. Specifically, to remove extraneous signal in the data while preserving model simplicity, we imposed a per cent variance cut-off of 1% as a criteria for selection of modes to include as features (figure 6). Note that only three modes are used in the analysis as cross-validation suggests these to be generalizable without over-fitting. The slow decay of the singular value spectrum is often handled by thresholding techniques which separate low-rank signal from noise [22]. The advantage of the three modes is also associated with interpretability and the visualization capabilities that three dimensions afford. Specifically, the figures presented show a clear pattern of clustering without producing high-dimensional space which is beyond our capabilities to visualize. The modal patterns associated with the three modes also allow for an interpretation of the time-frequency signatures that

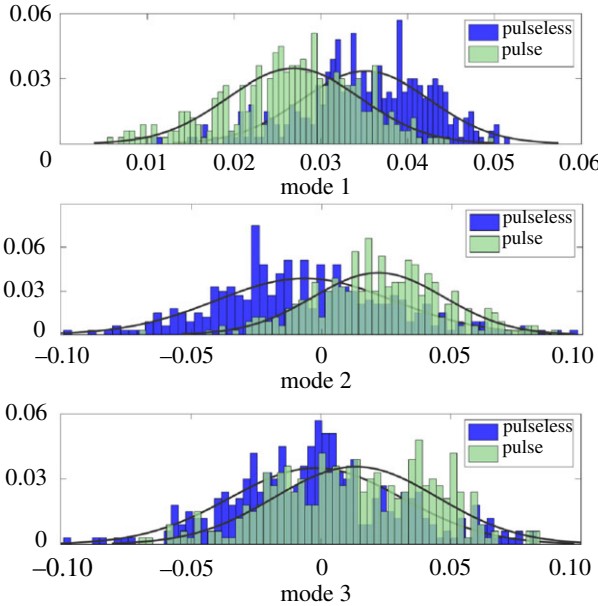

**Figure 7.** Histograms of Modes 1, 2 and 3 for each label during CPR using the training data.

dictate our ability to comprehend the ECG time-series recordings. In any case, cross-validation alone suggests that the three modes are appropriate to use for the clustering analysis that follows.

The first three modes each had greater than 1% variance, and together captured much of the variance of the system (figure 6). The distributions of Modes 1–3 in the ECG segments with CPR, stratified by pulse presence (figure 7) provided insight into their usefulness as features. Modes 1 and 2 had slightly overlapping, yet distinct distributions. While the distributions in Mode 3 showed more overlap, they were still slightly separated, indicating that the addition of Mode 3 possibly provided information that was not provided in Modes 1 and 2.

### 2.3.3. Clustering and classification

In order to classify ECG segments as pulse present or pulse absent, we considered the three modes derived from the PCA analysis, as well as a fourth variable, heart rate (in beats $min^{-1}$), as candidate features. Heart rate is a variable with accepted clinical importance, and to estimate heart rate, we first preprocessed the ECG with a higher-order (eighth order) Butterworth filter and a passband of 10–40 Hz to emphasize QRS complexes. We then applied a simple peak-finding algorithm to calculate heart rate from QRS complex locations observed within the ECG segment.

Using the training data, we explored a variety of classification models. In projecting the three PCA modes onto a three-dimensional space, we observed two distinct clusters: one with *Pulse* labels, the other with *No Pulse* labels. The formation of clusters in a low-dimensional feature space suggested amenability to discriminant analysis for which a number of classification models were suitable: linear discriminant analysis (LDA), quadratic discriminant analysis (QDA) and support vector machine (SVM) models. SVM model parameters were hand-tuned using a grid search [2,23]. These models fall under the broader aegis of supervised machine learning and are critical in automating data-driven discovery processes.

In addition to the above discriminant models, we also explored neural networks (NN), logistic regression (LR), Gaussian mixture models (GMM) and random forests (RF) with the aforementioned modes and heart rate as features. As we did not have enough samples for a deep NN, we used a two-layer NN to avoid overfitting, as there is much variability across patients. The number of trees in the RF algorithm was optimized using out-of-bag (OOB) error. Finally, as an alternative to classification involving the PCA modes, we used scalogram matrices as input images for a convolutional neural network (ConvNN). Convolutional neural networks are used heavily in image recognition and can sometimes detect nuances in images that are imperceptible to the human eye [20].

Separate classification models were fit for ECG segments with CPR and those without CPR. To select the optimal model, we assessed the performance of candidate models on training data by calculating the area under receiver operating characteristic curve (AUC) with a fivefold cross-validation procedure.

**Table 2.** Comparison of Training AUC values and 95% confidence intervals for candidate models and test performance for final model, where LDA, linear discriminant analysis; QDA, quadratic discriminant analysis; SVM, support vector machine; GMM, Gaussian mixture model; LR, logistic regression; RF, random forest; NN, neural network, ConvNN, convolutional neural network.

| | classifier | CPR | no CPR |
|---|---|---|---|
| **training** | | | |
| | LDA | 0.79 (0.76, 0.83) | 0.87 (0.84, 0.90) |
| | QDA | 0.80 (0.77, 0.84) | 0.87 (0.84, 0.93) |
| | SVM | 0.79 (0.75, 0.83) | 0.87 (0.84, 0.90) |
| | GMM | 0.77 (0.73, 0.81) | 0.85 (0.82, 0.88) |
| | LR | 0.79 (0.75, 0.83) | 0.86 (0.84, 0.89) |
| | RF | 0.80 (0.76, 0.83) | 0.87 (0.84, 0.90) |
| | NN | 0.80 (0.76, 0.83) | 0.87 (0.84, 0.89) |
| | ConvNN | 0.78 (0.75, 0.81) | 0.77 (0.74, 0.81) |
| **test** | | | |
| | LDA | 0.84 (0.80, 0.88) | 0.89 (0.86, 0.92) |

### 2.3.4. Assessment of classification performance

Based on the training performance, the best model was then tested on the external validation (test) set of ECG segments with and without CPR separately. Performance was characterized by AUC values, whose 95% confidence intervals were calculated using bootstrapping [24], and sensitivity and specificity for pulse detection. As there are not any clinically defined standards for sensitivity and specificity, we selected these values using the optimal operating point [25]. In addition, we compared this method with a previous algorithm which also classified organized rhythm segments during CPR [14]. That method used a logistic combination of three manually designed features: a measure of QRS rate and two measures of QRS amplitude and morphology (median magnitudes within two frequency bands centred at 6.25 and 25 Hz). AUCs of the two methods were compared with the Delong method for correlated data [26].

## 3. Results

Demographic, clinical and outcome characteristics of the study cohort are provided in table 1. There were 230 patients with 540 segment pairs (with and without CPR) in the training set and 153 patients with 372 segment pairs in the test set. The percentage of *Pulse* versus *No Pulse* segments were similar in the training (39% *Pulse*) and test (37% *Pulse*) sets.

We observed that the classifiers had comparable performance on the training data with and without CPR (table 2). While these models have similar AUC performance, they vary in complexity. Complex models like NN and ConvNN have a tendency to overfit and are less ideal due to the relatively small amount of data. As significantly more data is collected (specifically, by several orders of magnitude), it is expected that deep learning would provide a method to advance the state-of-the-art beyond the simpler regression methods presented. In order to reduce the risk of overfitting, we selected the simplest model, LDA, as the final model, which had an AUC of 79% during CPR and 87% without CPR (table 2). Furthermore, we observed that the addition of heart rate to PCA Modes 1–3 in the LDA model did not improve AUC, and therefore, the final model incorporates PCA Modes 1–3 as the sole features (electronic supplementary material, table S4).

AUC to discriminate between Pulse and *No Pulse* segments was 0.84 (95% CI (0.80, 0.88)) with CPR and 0.89 (95% CI (0.86, 0.92)) without CPR on test data. The sensitivity and specificity was 0.68 (95% CI (0.58, 0.77)) and 0.87 (95% CI (0.79, 0.93)) with CPR and 0.76 (95% CI (0.66, 0.84)) and 0.85 (95% CI (0.77, 0.91)) without CPR (electronic supplementary material, table S5).

When tested on the same dataset, this algorithm had marginally higher AUC values than the previous algorithm [14], but the AUC difference was not statistically significant ($p = 0.44$) during CPR. This algorithm performed significantly better than the previous algorithm without CPR ($p = 0.005$).

# 4. Discussion

An understanding of the ECG rhythm and its physiological consequences provides the basis for using ECG to predict pulse. The most common rhythms observed during cardiac arrest are ventricular fibrillation, asystole and organized rhythms. Ventricular fibrillation is characterized by chaotic, disorganized electrical activity on the ECG; and asystole by the absence of any electrical activity. Neither produces a pulse and their presence on the ECG obviates the need for a pulse check. Organized rhythms are defined by coordinated electrical activity on the ECG, but may or may not be associated with cardiac muscle contraction and a pulse (the latter commonly referred to as pulseless electrical activity). Thus when observed on the ECG, organized rhythms require a pulse assessment as to the continued need for CPR and resuscitative efforts.

A variety of modalities to detect spontaneous circulation during resuscitation have been investigated, but at present, automated tools to reliably identify a spontaneous pulse during CPR are not available [27]. The study of photoplethysmography is limited to pre-clinical data and pilot studies [28], and end-tidal carbon dioxide monitoring can be confounded by other interventions. Ultrasound is generally not available in the pre-hospital setting and can be operator dependent. ECG and thoracic impedance monitoring are universally available on defibrillators, and a few prior methods to automatically detect pulse have been reported [29–32]. However, those methods were not designed for real-time classification of pulse presence during CPR. In recent work, we developed an algorithm which predicts pulse presence in real time from the ECG during CPR among patients with underlying organized rhythms [14]. Ultimately, the optimal method to real-time assessment of a patient's physiological status and in turn the best choice of therapy may involve the integration of multiple technology-based measures. The ECG may be considered foundational given its universal use and fundamental role in resuscitation.

Although the method described in the current investigation and the recent prior method [14] were performed similarly during CPR, the current method integrates machine learning and signal processing techniques (figure 1) and presents a more scalable alternative to the recent prior method for pulse detection during CPR [14]. A limitation of the previous approach [14] was that it did not provide a framework to model more subtle morphologic characteristics nor to discover new feature spaces. By contrast, the components of the algorithm developed in the current method are modular and flexible; thus, new feature spaces and clustering techniques can be readily integrated and explored within the algorithmic structure proposed. As larger quantities of data become available, data-based approaches such as deep neural networks and more robust convolutional neural networks could easily be integrated into the current method to further improve performance. Importantly, the current algorithm can be applied in a real-time manner during resuscitation, as the signal processing and feature extraction steps are not computationally intensive. Specifically, wavelet transforms may be implemented using FFTs which modern embedded hardware is well-optimized to compute the transformations required. Computation time of PCA greatly relies on the length of the clip and the number of samples, both of which are relatively small. PCA is $O(mn^2 + n^3)$ [33], where $m$ is the length of the clip (2500 for CPR, 1250 for no CPR) and $n$ is the number of samples used. Given that we have small clips and relatively small sample sizes (compared with those used for NN), this method has the potential to be done in real time. In addition, the LDA model has already been trained, so features simply need to be projected on the LDA subspace to classify the rhythm (figure 8). This minimization of computational energy could allow the algorithm to be incorporated into devices currently in production and enable real-time feedback to rescuers during CPR. As this algorithm has the priority of scalability and in-field assessment, other manually designed and spectral features used in previous studies [29] were not used in this algorithm. The addition of such features could add computational complexity which would inevitably compromise scalability.

The algorithm described in the current study incorporates three spatial modes of the reduced wavelet scalogram, which enables some limited interpretability of the underlying variance of the two classes of data. Examination of the spatial modes demonstrates a band in most of the modes around approximately 1.5–2 Hz (figure 6). This fundamental band and its harmonics may represent some effects of CPR (guidelines recommend CPR rates of 100–120 compression per minute) [34], but could also represent underlying heart rate. Heart rate may be intrinsically incorporated in these three modes, which could account for why the addition of heart rate as a separate feature did not appreciably improve model performance versus use of Modes 1–3 alone. In Modes 1–3, there is also a significant amount of energy at higher frequencies. However, by contrast, Mode 4 exhibits minimal energy above the 2 Hz band other than the first and second harmonics (approx. 4 and 6 Hz). Because each mode represents a linear combination of the high-dimensional scalogram, there is not a definitive relationship between the

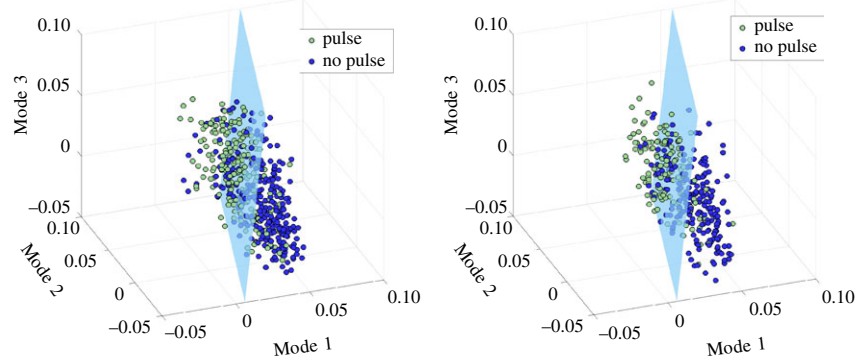

**Figure 8.** (*a*) Labelled training set with hyperplane. (*b*) Labelled test set with the same hyperplane.

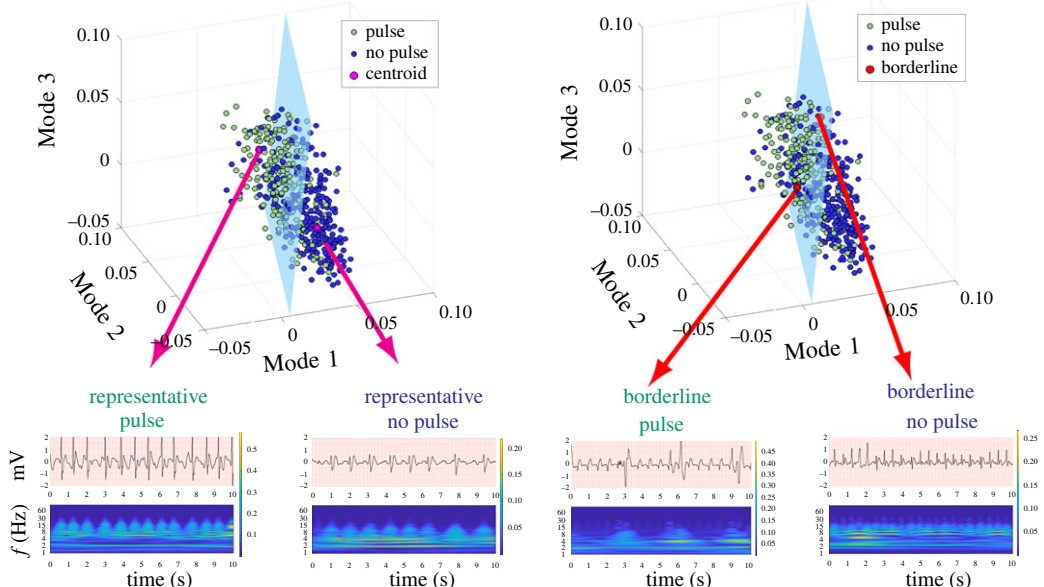

**Figure 9.** (*a*) Centroids of cluster for Pulse and No Pulse labels respectively, (*b*) borderline points for Pulse and No Pulse, respectively.

modes and conventionally understood ECG characteristics (e.g. amplitude, QRS width, QRS rate and rhythmicity). However, further insight can be gained by visualizing representative cases of the temporal modes of this analysis. For example, a case that maps to the cluster centroid of the *Pulse* classification has QRS complexes that are more narrow and larger in amplitude than the cluster centroid of the *No Pulse* classification, as well as a faster rate (figure 9). Narrower QRS complexes could be indicative of increased myocardial conduction velocity, and thus may potentially be represented by increased broad-band high-frequency content such as that observed in Modes 1–3 .

## 4.1. Limitations

In order to be deployed in clinical practice, the algorithm needs to be integrated with a function that can distinguish rhythms during CPR [35–37], because in application it assumes that the underlying rhythm is organized. In addition, further study is necessary to determine how this algorithm might be implemented into clinical practice to affect rescuer actions and resuscitation metrics such as CPR fraction and drug administration. Although the ascertainment of pulse presence used information from real-time audio and measured blood pressure, pulse detection can be clinically difficult and result in misclassification of its presence or absence that could attenuate the accuracy of the algorithm. While the study cohort included patients with initial rhythms of ventricular fibrillation, asystole and pulseless electrical activity, the initial rhythm distribution was weighted to include excess ventricular

fibrillation given the greater likelihood of spontaneous pulse and the added analytical power. Moreover, cases came from a single EMS system with generally good outcomes. These characteristics (initial rhythm distribution and EMS system) may affect algorithm performance and hence the generalizability of the results.

## 4.2. Future directions

There are many ongoing challenges and promising directions that motivate future research in this area. The raw ECG data-streams remain noisy and have frequent artefacts. Machine learning architectures can potentially be integrated at the front end of methods in order to help provide more robust filtering of the ECG time-series data. Furthermore, as recording technologies improve and more data is available, more powerful neural network architectures can be used in the classification step to drive accuracies even higher. Although the wavelet decomposition proved to be highly advantageous for extracting key features of the pulse presence, there is a potential for improving the feature space used for extraction of dominant spatio-temporal signals responsible for accurate clustering and thus improving model performance. Regardless, the data-driven discovery pipeline proposed here provides a baseline model for which improvements can be readily integrated and executed.

# 5. Conclusion

In summary, we developed a machine learning framework for automating the accurate prediction of pulse presence from the ECG time series. The proposed algorithmic architecture integrates and leverages three mathematical architectures: (i) time-series analysis through a wavelet decomposition, (ii) feature engineering through dominant correlated spatio-temporal patterns, and (iii) clustering and classification methods. We demonstrated moderate predictive ability in identifying pulse presence even while CPR is being performed on a patient, which has potential to provide essential feedback to rescuers during uninterrupted CPR. The mathematical approach provides a data-driven framework that can be applied widely across current technology, giving a generalizable approach for clinical pulse prediction to potentially improve outcomes following cardiac arrest. Indeed, this approach suggests an opportunity to improve resuscitation through real-time identification of pulse presence to minimize CPR interruptions and inform treatment decisions.

Data accessibility. The following algorithm was implemented on Matlab (R2018b). The code and data can be accessed using the following link: https://github.com/dsashid/PulsePrediction [38]. Data repository can be found using this link: https://doi.org/10.5281/zenodo.3995071. Data are in the form of ECG scalograms with and without CPR and can be imported directly into the code.
Authors' contributions. D.S. conducted data analysis, implemented the method, wrote code and drafted the original manuscript. H.K. helped with the statistical analyses, implementation of method and critically revised the manuscript. J.C. helped with the statistical analyses, implementation of method and critically revised the manuscript. J.B. carried out data handling, pre-processing and creation of the demographics table (table 1). S.B. carried out the pulse annotations and labelling of data. P.J.K. participated in the design of the study, offered valuable clinical input and interpretation, and revised the manuscript; T.D.R. conceived of the study, coordinated the study and drafted the manuscript. J.N.K. advised on the design of the method, coordinated the study and drafted the manuscript. All authors gave final approval for publication and agreed to be held accountable for the work performed therein.
Competing interests. We declare we have no competing interests.
Funding. This study was supported in part by grants provided to the University of Washington by the American Heart Association (D.S., H.K., T.D.R. and P.J.K.) and the Washington Research Foundation (J.C.).
Acknowledgements. We appreciate the emergency medical services personnel of King County, without whom this study would not have been possible. Informative discussions on pulse detection were conducted with Dawn Jorgensen, Stacy Gehman and Chenguang Liu (Philips Healthcare, Bothell, WA).

# Appendix A

In order to assess the ideal wavelet transform for feature extraction, we use a subset of the dataset with 120 ECG strips (96 training) with the pulse label and 120 (96 training) ECG strips with the pulseless label. For each of these wavelets, we examined various combinations of the modes and discriminants and generated the output of the combination with the highest AUC using this subset of the training data.

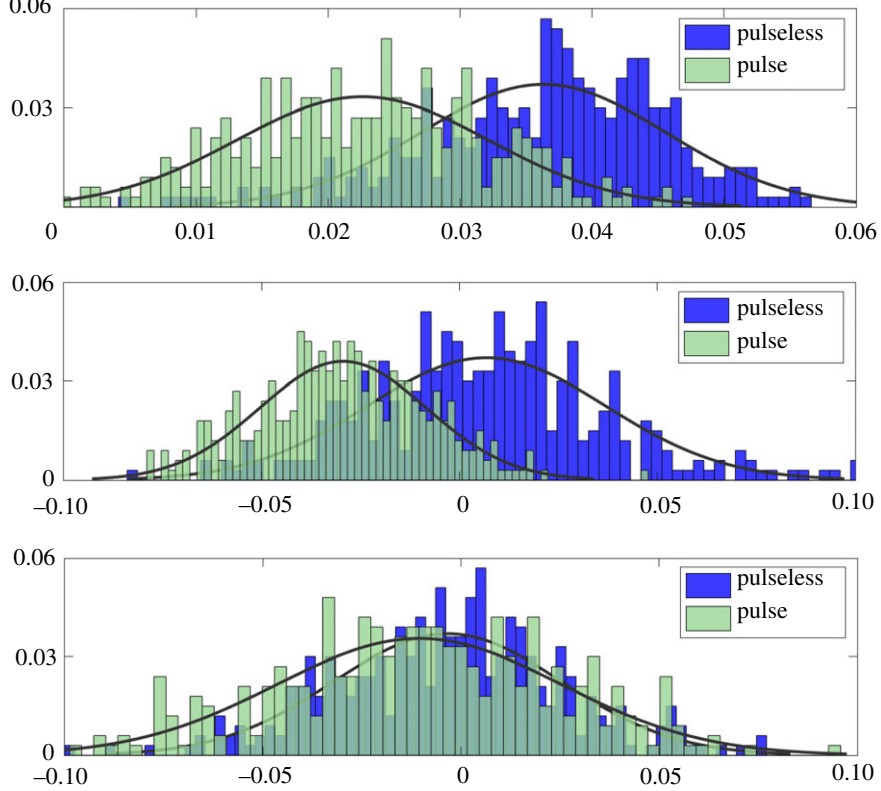

**Figure 10.** Histograms of Modes 1, 2 and 3 without CPR using training data.

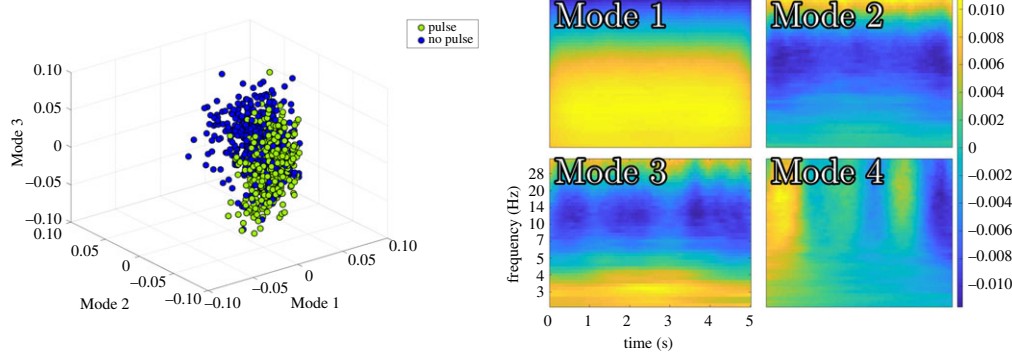

**Figure 11.** Temporal and spatial modes for PCA of reshaped scalograms (without CPR).

To find the optimal model, we examined various discriminant models, neural networks and RF models. The final model was selected based on a combination of training AUC, sensitivity and specificity. These model performances are shown in electronic material, table S5. The highlighted row is the final model performance when applied to test data.

In addition to creating a model for CPR-artefacted segments, we created a model for segments without CPR. Figure 10 shows the histograms for segments (without CPR) with 'Pulse' and 'No Pulse' outcomes. As seen in the CPR-artefacted segments, the distributions for Modes 1 and 2 (without CPR) are distinct. Figure 11 shows the first three temporal modes when projected onto a three-dimensional space on the left and the first four spatial modes of reshaped scalograms (without CPR) on the right.

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
