## [Peer Review File · Royal Society Open Science]

Review History

RSOS-210566.R0 (Original submission)

Review form: Reviewer 1

Is the manuscript scientifically sound in its present form?

Yes

Are the interpretations and conclusions justified by the results?

Yes

Is the language acceptable?

Yes

Do you have any ethical concerns with this paper?

Yes

Have you any concerns about statistical analyses in this paper?

No

Recommendation?

Accept as is

Comments to the Author(s)

Thank you for addressing issues flagged in the first round of reviews. It seems that many of the reviewer concerns were driven by sub-optimal reporting, which has now been addressed

Review form: Reviewer 2

Is the manuscript scientifically sound in its present form?

No

Are the interpretations and conclusions justified by the results?

No

Is the language acceptable?

Yes

Do you have any ethical concerns with this paper?

No

Have you any concerns about statistical analyses in this paper?

Yes

Recommendation?

Reject

Comments to the Author(s)

See appendices A & B.

Review form: Reviewer 3

Is the manuscript scientifically sound in its present form?

Yes

Are the interpretations and conclusions justified by the results?

Yes

Is the language acceptable?

Yes

Do you have any ethical concerns with this paper?

No

Have you any concerns about statistical analyses in this paper?

No

Recommendation?

Accept with minor revision (please list in comments)

Comments to the Author(s)

The manuscript describes and aims to provide a solution to a very relevant problem facing medical community. It is well written and I enjoyed reading it. Since the contributions are significant, I recommend publication after authors address following questions:

1. On page 2, authors state that they worked with two separate data sets, one sampled at 125 Hz and another at 250 Hz. Authors should comment on how their model trained on one kind of data, could (could not) be applied to the real world data which may have different sampling rates/ underlying distributions.
2. Authors state that first 3 modes of PCA are chosen that capture most of the variance. Generally it's recommended to choose number of modes that capture around 95% of the variance. From figure 6 it seems that the selected 3 modes only represent about 10% of the variance. If that is the case, selected modes that are fed to the classification model may not accurately represent the underlying data.
3. The authors state that their method is scalable and preprocessing calculations to generate features can be done in real time. Can authors comment on the time complexity of these calculations.
4. Wouldn't directly feeding ECG data to a deep neural network such as RNN be faster, and neural network can figure out the underlying relevant features? Please comment.

Decision letter (RSOS-210566.R0)

Dear Ms Sashidhar

The Editors assigned to your paper RSOS-210566 "Machine Learning and Feature Engineering for Predicting Pulse Presence during Chest Compressions" have now received comments from reviewers and would like you to revise the paper in accordance with the reviewer comments and any comments from the Editors. Please note this decision does not guarantee eventual acceptance.

We do not generally allow multiple rounds of revision so we urge you to make every effort to fully address all of the comments at this stage. If deemed necessary by the Editors, your

manuscript will be sent back to one or more of the original reviewers for assessment. If the original reviewers are not available, we may invite new reviewers.

Please submit your revised manuscript and required files (see below) no later than 21 days from today's (ie 20-Jul-2021) date. Note: the ScholarOne system will 'lock' if submission of the revision is attempted 21 or more days after the deadline. If you do not think you will be able to meet this deadline please contact the editorial office immediately.

on behalf of Prof R. Kerry Rowe (Subject Editor)
openscience@royalsociety.org

Associate Editor Comments to Author:

Thank you for engaging with the reviewers' comments. There remains a split between recommendations to publish or reject. Given the relatively positive comments from the third reviewer, we would like you to revise the paper to address the concerns raised by two of the referees. Please carefully revise the paper to tackle these concerns and provide a point-by-point response to highlight the changes made. Good luck!

Reviewer comments to Author:

Reviewer: 1

Comments to the Author(s)

Thank you for addressing issues flagged in the first round of reviews. It seems that many of the reviewer concerns were driven by sub-optimal reporting, which has now been addressed

Reviewer: 2

Comments to the Author(s)

see attached

Reviewer: 3

Comments to the Author(s)

The manuscript describes and aims to provide a solution to a very relevant problem facing medical community. It is well written and I enjoyed reading it. Since the contributions are significant, I recommend publication after authors address following questions:

1. On page 2, authors state that they worked with two separate data sets, one sampled at 125 Hz and another at 250 Hz. Authors should comment on how their model trained on one kind of data, could (could not) be applied to the real world data which may have different sampling rates/ underlying distributions.
2. Authors state that first 3 modes of PCA are chosen that capture most of the variance. Generally it's recommended to choose number of modes that capture around 95% of the variance. From figure 6 it seems that the selected 3 modes only represent about 10% of the variance. If that is the case, selected modes that are fed to the classification model may not accurately represent the underlying data.
3. The authors state that their method is scalable and preprocessing calculations to generate features can be done in real time. Can authors comment on the time complexity of these calculations.
4. Wouldn't directly feeding ECG data to a deep neural network such as RNN be faster, and neural network can figure out the underlying relevant features? Please comment.

===PREPARING YOUR MANUSCRIPT===

===PREPARING YOUR REVISION IN SCHOLARONE===

To revise your manuscript, log into <https://mc.manuscriptcentral.com/rsos> and enter your Author Centre - this may be accessed by clicking on "Author" in the dark toolbar at the top of the

page (just below the journal name). You will find your manuscript listed under "Manuscripts with Decisions". Under "Actions", click on "Create a Revision".

Author's Response to Decision Letter for (RSOS-210566.R0)

See Appendix C.

RSOS-210566.R1 (Revision)

Review form: Reviewer 3

Is the manuscript scientifically sound in its present form?

Yes

Are the interpretations and conclusions justified by the results?

Yes

Is the language acceptable?

Yes

Do you have any ethical concerns with this paper?

No

Have you any concerns about statistical analyses in this paper?

No

Recommendation?

Accept as is

Comments to the Author(s)

Authors have answered my questions sufficiently and I recommend accepting the manuscript for publication.

Decision letter (RSOS-210566.R1)

Dear Ms Sashidhar

On behalf of the Editors, we are pleased to inform you that your Manuscript RSOS-210566.R1 "Machine Learning and Feature Engineering for Predicting Pulse Presence during Chest Compressions" has been accepted for publication in Royal Society Open Science subject to minor revision in accordance with the referees' reports. Please find the referees' comments along with any feedback from the Editors below my signature.

We invite you to respond to the comments and revise your manuscript. Below the referees' and Editors' comments (where applicable) we provide additional requirements. Final acceptance of

your manuscript is dependent on these requirements being met. We provide guidance below to help you prepare your revision.

Please submit your revised manuscript and required files (see below) no later than 7 days from today's (ie 01-Oct-2021) date. Note: the ScholarOne system will 'lock' if submission of the revision is attempted 7 or more days after the deadline. If you do not think you will be able to meet this deadline please contact the editorial office immediately.

on behalf of R. Kerry Rowe (Subject Editor)
openscience@royalsociety.org

Associate Editor Comments to Author:

Comments to the Author:

Thank you for your patience - unfortunately, one of the previous reviewers was unavailable for this revision, and (though we've sought alternatives) the editors have decided to make a decision based on the commentary that has been received thus far. In essence, the view is that the paper may be accepted with minor revisions - in this case, the specific revision regards the ethical approvals sought to support this study.

It has been noted that, between the initial submission and revision, you added a note that approvals were sought and granted, but you have not provided specific details regarding those approvals. Please can you ensure that you amend the ethical approval statement to include not only the details of the awarding panel(s) but the licence/appoval/permit numbers you were provided with, too? In the (seemingly unlikely) event that none were provided, please clearly state this and - ideally - include the contact details for the approval bodies, so interested readers can explore this further.

Reviewer comments to Author:

Reviewer: 3

Comments to the Author(s)

Authors have answered my questions sufficiently and I recommend accepting the manuscript for publication.

===PREPARING YOUR MANUSCRIPT===

one version identifying all the changes that have been made (for instance, in coloured highlight, in bold text, or tracked changes);
 a 'clean' version of the new manuscript that incorporates the changes made, but does not highlight them. This version will be used for typesetting.

===PREPARING YOUR REVISION IN SCHOLARONE===

- Any electronic supplementary material (ESM).
- If you are requesting a discretionary waiver for the article processing charge, the waiver form must be included at this step.
- If you are providing image files for potential cover images, please upload these at this step, and inform the editorial office you have done so. You must hold the copyright to any image provided.
- A copy of your point-by-point response to referees and Editors. This will expedite the preparation of your proof.

- Ensure that your data access statement meets the requirements at <https://royalsociety.org/journals/authors/author-guidelines/#data>. You should ensure that you cite the dataset in your reference list. If you have deposited data etc in the Dryad repository, please only include the 'For publication' link at this stage. You should remove the 'For review' link.
- If you are requesting an article processing charge waiver, you must select the relevant waiver option (if requesting a discretionary waiver, the form should have been uploaded at Step 3 'File upload' above).
- If you have uploaded ESM files, please ensure you follow the guidance at <https://royalsociety.org/journals/authors/author-guidelines/#supplementary-material> to include a suitable title and informative caption. An example of appropriate titling and captioning may be found at https://figshare.com/articles/Table_S2_from_Is_there_a_trade-off_between_peak_performance_and_performance_breadth_across_temperatures_for_aerobic_scope_in_teleost_fishes_/3843624.

Author's Response to Decision Letter for (RSOS-210566.R1)

See Appendix D.

Decision letter (RSOS-210566.R2)

Dear Ms Sashidhar,

I am pleased to inform you that your manuscript entitled "Machine Learning and Feature Engineering for Predicting Pulse Presence during Chest Compressions" is now accepted for publication in Royal Society Open Science.

If you have not already done so, please remember to make any data sets or code libraries 'live' prior to publication, and update any links as needed when you receive a proof to check - for

instance, from a private 'for review' URL to a publicly accessible 'for publication' URL. It is good practice to also add data sets, code and other digital materials to your reference list.

Appendix A

RESPONSE TO THE REVISION:

The authors have attempted to address reviewers' comments to provide better clarification. [We noticed that both reviewers expressed concern concerning the initial cardiac arrest of the patients.] Although it is much clearer what parts of the ECG recordings the authors used, a more detailed description of the data is still missing.

In summary, I cannot recommend the publication of this manuscript.

Here are some comments the author may find helpful if they wish to submit the paper elsewhere.

The description of the wavelet transform and the principal component analysis is very detailed and provides maybe somewhat unnecessary or trivial details. In comparison, the data description that is eventually used in training and testing machine learning algorithm is limited to one paragraph in the results section (p.6, l.46-51). A table similar to Table 1 that describes the demographics of the patient should provide similar details of the final data used, including numbers (not just percentages) of the segments used in each category mentioned in the paragraph on p.6, l. 46-51.

It is still not very clear what defines an organized rhythm in the ECG signal. An illustration or a short description would clarify the misunderstanding between organized rhythm and pulse rhythm. Although the authors provide the initial cardiac arrest rhythm details (VF, PEA, Asystole) in the Methods section (p.2, l.38), it seems that at the end, the data used for the machine learning algorithm is only divided into two pulse rhythms, pulse or PEA: "either pulse present or absent (i.e., PEA)," p.2,l.51. Thus, a clarification here and in the results section (p.6,l.46-51) is crucial to properly interpret the obtained results of this study.

The ECG segments were divided with respect to their patients when the data was divided into training and testing data sets. This seems somewhat arbitrary and could actually introduce a bias that could stem from the patients. Assigning the segments to the data sets randomly should improve the generalization of the classifier and the testing results.

It is not clear how the pulse can be calculated from the QRS complex in the ECG segment of a pulseless segment.

Appendix B

This is indeed a timely study to address a long-standing problem from a medical perspective. The described methods to i) extract time-frequency characteristics of the ECG signal using wavelet transform (WT) and ii) incorporate principle component analysis (PCA) to select the "most relevant" frequency bandwidths are not new by themselves but are cleverly used in this application and deserve recognition. The methodology section is well written except for some minor structural issues.

However, there are some significant issues with the design of the study and interpretation of the results. The following comments outline these critical design issues and provide some feedback for the structure of the paper.

Because of the outlined flaws in the experiment's design, I regret that I cannot recommend this paper for publication.

P. 2, L. 32 or P.7 Table I: Why were VF patients included in this study? As the authors explain themselves in the Discussion Section (p.6, l.52), arrests such as VF and Asystole are two conditions that are very different from the PEA arrests. These arrests do not require pulse assessment during CCs, since the only time the pulse will reappear is right after successful defibrillation followed by ROSC. It should be noted here that the guidelines for an EMT state that no CCs should be performed during the defibrillation. Thus, the VF patient should have been excluded from this investigation.

With roughly 87% of the included patients in this study having an arrest due to VF or Asystole, this study presents a massive flaw in its design, making the obtained results invalid and eliminating several derived conclusions mentioned in the discussion section.

Here are some specific issues with the methods and the results.

- The classifiers' performance would be higher during No CPR segments, and misclassification would be anticipated due to the VF segments.
- Similar performance of the classifiers, after the inclusion of the calculated HR as a feature, is likely due to the VF segments.
- Asystole patients do not have any electrical activity, whether experiencing CC or not, unless the defibrillation was performed successfully and the pulse restored right after it.
- The developed algorithm to calculate the HR, as described on page 4 and line 49, cannot be applied to the VF segments because the 'QRS complexes' that the authors are using is not present in the chaotic VF ECG waveform.
- The developed algorithm to calculate the HR cannot be accepted as a norm and assumed to be 100% accurate. An independent signal such as a SpO2 reading should be used as the norm to confirm the algorithm's accuracy or used as the HR feature (4th feature) right away.

Some further comments are as follows

- P. 1, L. 33: Perhaps an illustration of the chest compression (CC) interference would be of help and can be shown as an example plot in Fig. 2 instead of the current plot.

- P. 2, L. 21: Filtering is an unnecessary step in this analysis since the bandwidth of the described WT can be limited by the ' a ' variable in equation (1).
- P. 3, L. 1: Figure 3 should be eliminated since it is never referenced in the text.
- P. 4, L. 42: Figure 5 should be eliminated since it is never referenced in the text and shows almost the same idea as Figure 4.
- P. 6, L. 47: The paragraph stating the number of patients and the number of segments used, and the population concerning the output (pulse vs. no pulse) should be moved to the data (methodology) section.
- The authors should avoid using expressions such as: "detected heart rate" "pulse status," which implies detection or status of the heart rate value as well, and should use "pulse occurrence" "pulse presence" instead.

Appendix C

Reviewer: 1

Comments to the Author(s)

Thank you for addressing issues flagged in the first round of reviews. It seems that many of the reviewer concerns were driven by sub-optimal reporting, which has now been addressed

We thank the reviewer for the comments and taking the time to look at our manuscript.

Reviewer 2:

The description of the wavelet transform and the principal component analysis is very detailed and provides maybe somewhat unnecessary or trivial details. In comparison, the data description that is eventually used in training and testing machine learning algorithm is limited to one paragraph in the results section (p.6, l.46-51). A table similar to Table 1 that describes the demographics of the patient should provide similar details of the final data used, including numbers (not just percentages) of the segments used in each category mentioned in the paragraph on p.6, l. 46-51.

Study data are presented on a patient level for patients in the training, test, and overall groups to allow comparison of the dataset. We concur that this is a fairly granular presentation of the dataset demographics but feel that this allows comparison of the training and test randomization. Unfortunately, it is not possible to report patient-level demographics for the pulse and pulseless class subgroups, because of their mix of patients – i.e. each patient may have had up to a predefined limit of three ECG samples from each of the two pulse or no pulse classes. Instead we have added the median number of ECG samples each patient provided to the training and test sets so the reader has this insight.

The following was added to the manuscript:

“Table 1 also shows the median number of samples (including pulse and pulseless) for each patient for training and test sets respectively.”

It is still not very clear what defines an organized rhythm in the ECG signal. An illustration or a short description would clarify the misunderstanding between organized rhythm and pulse rhythm. Although the authors provide the initial cardiac arrest rhythm details (VF, PEA, Asystole) in the Methods section (p.2, l.38), it seems that at the end, the data used for the machine learning algorithm is only divided into two pulse rhythms, pulse or PEA: "either pulse present or absent (i.e., PEA)," p.2,l.51. Thus, a clarification here and in the results section (p.6,l.46-51) is crucial to properly interpret the obtained results of this study.

We agree that the description of rhythm and pulse categories could be improved. In the case of cardiac arrest, three ECG states that are commonly observed dynamically throughout resuscitation are ventricular fibrillation, asystole, and organized. In our Table 1, as is common with EMS systems, patients are generally categorized into groups depending on the initial rhythm observed by EMS personnel, but the rhythm can change throughout the resuscitation. An organized rhythm is defined as discrete repetitive electrical complexes on the ECG (indicative of ventricular depolarization) at a rate that would be expected to produce a discernible pulse, but may or may not do so. In the absence of a mechanical pulse, an organized ventricular rhythm is referred to as pulseless electrical activity. Figure 4 provides two representative examples of the organized rhythm categories used in this study; one that resulted in mechanical pulse, and one that did not. Other rhythm categories which may be observed during cardiac arrest (e.g. asystole and ventricular fibrillation) do not produce a pulse. Hence the study only includes ECG segments during organized rhythm (the only category of the 3 that might be expected to produce a mechanical pulse) in order to predict those with and without pulse. This is highlighted in the methods indicating that data were included only during organized rhythm as follows,

"We developed and evaluated the algorithm on adjacent ECG segments at each pulse assessment with an organized rhythm, as creation of a pulse is only possible from an organized rhythm. Specifically, for each pulse check, a 10-second organized rhythm ECG segment was collected during the period of ongoing CPR just prior to the pulse check,..."

The following definition was added to the manuscript:

"An organized rhythm is defined as discrete repetitive electrical complexes on the ECG (indicative of ventricular depolarization) at a rate that would be expected to produce a discernible pulse, but may or may not do so."

The ECG segments were divided with respect to their patients when the data was divided into training and testing data sets. This seems somewhat arbitrary and could actually introduce a bias that could stem from the patients. Assigning the segments to the data sets randomly should improve the generalization of the classifier and the testing results.

We appreciate the suggested alternate approach that would randomize based on the individual clip, rather than at the patient level. The reason we chose instead to randomize the data on the patient level was in order to truly test generalizability of the method to unique patients. Otherwise, if data from the same patients were present in both the training and test, the clinical generalizability may be less certain, as data within each patient may be correlated. To control for potential bias caused by overrepresentation of any single patient, we only included a maximum of three samples per class from any one patient.

Explaining the rationale behind our method of rationalization is important, and we thank the reviewer for raising the issue.

Accordingly, the following was added to the manuscript:

“Note that we have randomized the data on a patient level in order to test generalizability to patients in a withheld test set.”

It is not clear how the pulse can be calculated from the QRS complex in the ECG segment of a pulseless segment.

We believe the reviewer is referring to how the heart rate was calculated from the QRS complexes in the ECG segment of a pulseless segment (i.e. the heart rate of a pulseless electrical activity rhythm). The heart rate was calculated by dividing the number of successive QRS complexes in that clip by the duration of that clip (e.g. 5 complexes within a clip of 6 seconds duration would correspond to an average rate of 50 complexes per minute). This rate corresponded to the electrical rate of the rhythm, irrespective of whether or not it generated a pulse.

We have added ‘mechanical pulse’ to the manuscript to clarify:

Organized rhythms were further categorized as either **mechanical** pulse present or absent (i.e., pulseless electrical activity) by reviewing the audio recording, as EMS rescuers verbalize the results of manual pulse assessment and confirm any positive pulse assessment by blood pressure measurement per protocol.

Ideally, the rhythm and presence of a **mechanical** pulse would be monitored continuously during resuscitation to help guide treatment decisions.

Reviewer: 3

Comments to the Author(s)

The manuscript describes and aims to provide a solution to a very relevant problem facing medical community. It is well written and I enjoyed reading it. Since the contributions are significant, I recommend publication after authors address following questions:

1. On page 2, authors state that they worked with two separate data sets, one sampled at 125 Hz and another at 250 Hz. Authors should comment on how their model trained on one kind of data, could (could not) be applied to the real world data which may have different sampling rates/ underlying distributions.

The reviewer is correct in pointing out that time-series data were originally collected at sampling rates between 125-250 Hz (specifically 125, 200, and 250 Hz). These rates represent the

sampling rates for three commonly used automated external defibrillator models, all of which have been homogenized to 250 Hz. We have added the following sentences to clarify:

“To ensure that the underlying method would be applicable across this range in sampling rates, all data were low-pass filtered at 40 Hz after resampling all data to a common sampling rate of 250 Hz.

Since the lowest original sampling rate is 125 Hz (with a Nyquist frequency of 62.5Hz), we assume that the 40 Hz cutoff frequency assures the same maximum frequency content across all devices, eliminating the possibility that the algorithm would utilize frequencies above 40 Hz that may not be as well-resolved in devices with lower sampling rates.”

2. Authors state that first 3 modes of PCA are chosen that capture most of the variance. Generally it's recommended to choose number of modes that capture around 95% of the variance. From figure 6 it seems that the selected 3 modes only represent about 10% of the variance. If that is the case, selected modes that are fed to the classification model may not accurately represent the underlying data.

Thank you for your response. We agree with the reviewer that ideally we would want the modes to capture around 95% of the variance. However, in cases with high corruption and noise (such as this), this may not always be the case. In our study, specifically, we were working with a 200 dimensional set with a very “long tail.” In other words, while monotonically decreasing, the modes following mode 3 carried a minute amount of variance (each mode $\ll 1\%$). Thus, given that the first three modes were still significantly larger than the rest of the modes, we chose a 1% cutoff.

The mode selection was backed up by cross-validation, or hyper parameter tuning. That is, most of the time in selecting the number of modes to use, it is checked against cross-validation to be sure to not include too many extraneous modes. This has been a long-standing issue with PCA which only recently has been addressed with theoretical foundations (See Gavish & Donoho 2014). A smaller amount of modes is generally preferred to a larger number since it typically improves generalization errors and circumvents over-fitting. The smaller number of modes also improves the information criteria such as AIC (Akaike information criteria) and BIC (Bayesian information criteria) since best-fit models are penalized by the number of terms used versus the reconstruction/classification accuracy. In our case, there is also the added advantage of being able to visualize the modes and their classification capabilities in 3D plots. We feel this is a nice step towards interpretability since the separation of data isn't happening in some high-dimensional space beyond our capabilities to understand or visualize.

Of course, we would gladly use more modes if cross-validation suggested it. But the three modes used are quite strong in the performance and generalize quite well (both of which are tested via cross-validation).

To address this in the manuscript, we have added the following:

“Note that only three modes are used in the analysis as cross-validation suggests these to be generalizable without over-fitting. The slow decay of the singular value spectrum is often handled by thresholding techniques which separate low-rank signal from noise~\cite{gavish2014optimal}. The advantage of the three modes is also associated with interpretability and the visualization capabilities that three dimensions afford. Specifically, the figures presented show a clear pattern of clustering without producing high-dimensional space which is beyond our capabilities to visualize. The modal patterns associated with the three modes also allow for an interpretation of the time-frequency signatures that dictate our ability to comprehend the ECG time-series recordings. In any case, cross-validation alone suggest that the three modes are appropriate to use for the clustering analysis that follows.”

3. The authors state that their method is scalable and preprocessing calculations to generate features can be done in real time. Can authors comment on the time complexity of these calculations.

Wavelet transforms may be implemented using FFTs which modern embedded hardware is well-optimized to compute. PCA is $O(m^2n + mn^2)$, where m is the length of the clip (2500 for CPR, 1250 for no CPR) and n is the number of samples used. Given that we have small clips and relatively small sample sizes (compared to those used for NN), this method has the potential to be done in real time. In addition, the LDA model has already been trained, so features simply need to be projected on the LDA subspace to classify the rhythm.

The following sentence has been added:

“Specifically, wavelet transforms may be implemented using FFTs which modern embedded hardware is well-optimized to compute the transformations required. Computation time of PCA greatly relies on the length of the clip and the number of samples, both of which are relatively small. PCA is $O(m^2n + mn^2)$, where m is the length of the clip (2500 for CPR, 1250 for no CPR) and n is the number of samples used. Given that we have small clips and relatively small sample sizes (compared to those used for NN), this method has the potential to be done in real time. In addition, the LDA model has already been trained, so features simply need to be projected on the LDA subspace to classify the rhythm.”

4. Wouldn't directly feeding ECG data to a deep neural network such as RNN be faster, and neural network can figure out the underlying relevant features? Please comment.

Due to the difficulty in obtaining real-world out-of-hospital cardiac arrest data, there were insufficient data in our study to properly train a deep network approach. Deep neural networks require significant data in order to be effective, as was historically shown in the famous 2014 ImageNet problem where deep neural networks were first shown to be effective at scale. In this case, ImageNet provided orders of magnitude more data than had been previously used to train a neural network, allowing it to be successful. (See “Deep Learning” by Lecun, Bengio and Hinton, Nature 2015 for an analysis of the data required for making deep learning successful). Data do not currently exist anywhere near that required to allow deep learning to excel above other methods. As shown in Table III, we ran a Convolutional Neural Network (Table III) on the scalograms as well. The classifier had the following performances for CPR and No CPR, respectively: 0.78 (0.75,0.81) and 0.77 (0.74,0.81), both of which are lower than those of the LDA model. Similarly given the amount of noise found in the data, we would need many more samples for a neural network to work well. Given this lack of data, we turn to wavelet transform and dimensionality reduction (through PCA) to separate potential noise and discover underlying features.

This issue is highlighted in the Results section of the paper. We have added the following sentence to elaborate.

“Complex models like Neural Networks and Convolutional Neural Networks have a tendency to overfit and are less ideal due to the relatively small amount of data. **As significantly more data is collected (specifically, by several orders of magnitude), it is expected that deep learning would provide a method to advance the state-of-the-art beyond the simpler regression methods presented.**”

This importance of this feature extraction is highlighted in the Methods section of the paper:

“Since wavelets can extract localized features in the time-frequency domain, we applied the wavelet transforms to the detrended ECG time-series measurements.”

Appendix D

Comment to the authors:

It has been noted that, between the initial submission and revision, you added a note that approvals were sought and granted, but you have not provided specific details regarding those approvals. Please can you ensure that you amend the ethical approval statement to include not only the details of the awarding panel(s) but the licence/approval/permit numbers you were provided with, too? In the (seemingly unlikely) event that none were provided, please clearly state this and - ideally - include the contact details for the approval bodies, so interested readers can explore this further.

We thank the editor for their comments. We have added the following sentence to denote approval number and details regarding the review committee. Any other questions regarding this study can be sent to the corresponding author's email (dsashid@uw.edu) which is listed on the footnote of the first page.

The study was reviewed and approved by the Human Subjects Review Committee of the University of Washington who granted exemption from informed consent under established minimal risk criteria in compliance with applicable government regulations (STUDY00013007), with concurrence by the Research Oversight Review Committee of Public Health – Seattle and King County who ceded their approval to the University of Washington.

Reviewer comments to Author:

Reviewer: 3

Comments to the Author(s)

Authors have answered my questions sufficiently and I recommend accepting the manuscript for publication.

We thank the reviewer for their comment and acceptance of the manuscript.